# An E-Learning Program for Continuing Midwifery Education on Handling High-Risk Abuse Cases: A Pretest–Posttest Design

**DOI:** 10.3390/ijerph20136317

**Published:** 2023-07-07

**Authors:** Kaori Baba, Yaeko Kataoka

**Affiliations:** 1Research Center for Social Science & Medicine, Tokyo Metropolitan Institute of Medical Science, Tokyo 156-8506, Japan; 2Graduate School of Nursing Science, St Luke’s International University, Tokyo 104-0044, Japan; yaeko-kataoka@slcn.ac.jp

**Keywords:** education, child abuse, midwifery, nurses

## Abstract

It is essential to equip midwives and nurses working in the perinatal period with comprehensive knowledge and awareness regarding child abuse prevention. However, most midwives and nurses in Japan do not have the opportunity to learn about abuse prevention during their basic education. We aimed to develop an e-learning program to assist obstetric midwives and nurses in acquiring the knowledge needed to provide support and handle cases with a high risk of abuse, as well as to assess the program’s usefulness. This study employed a single-group pre–post design; e-learning served as the intervention. Seventy-one obstetric midwives and nurses were recruited. The program’s usefulness was the difference between the participants’ pretest and post-test knowledge and efficacy scores. The score data were analyzed using the t-test. A paired t-test revealed that the post-test scores of knowledge and efficacy were significantly higher than those of the pretest, with a large effect size (d = 1.71). Platforms where basic knowledge on how to respond to high-risk abuse cases are lacking in nursing education; thus, this e-learning program is recommended for nurses working in the perinatal field throughout Japan. This educational opportunity for perinatal midwives and nurses will increase awareness and contribute to abuse prevention.

## 1. Introduction

Child abuse is a global social issue, and within the realm of perinatal nursing, supporting caregivers to prevent subsequent abuse is a critical responsibility. The effectiveness of programs for tertiary prevention intervention for abuse has not been established, as demonstrated by previous studies [1,2]. Hence, Japan has recently recognized the importance of primary prevention during pregnancy [3,4]. That pregnant women should be prioritized for support [1] is supported by national surveys in Japan, which indicate that 75 percent of abuse deaths occurred less than three months after childbirth and that mothers are the majority of abusers [5,6]. Primary prevention in the perinatal period will enable the avoidance of various future risks of abuse, including adverse effects on the brain [7], increased susceptibility to post-traumatic stress disorder [8], and an increased likelihood of developing other psychotic disorders [9], with the damage extending into adulthood [10,11].

At the same time, there is little education or guidance related to abuse prevention targeted for perinatal nurses [12], and Japan is no exception. Japan’s current basic nursing and midwifery education does not include curricula for addressing abuse. Hence, most obstetric nurses do not have the opportunity to acquire sufficient knowledge regarding abuse prevention through their basic nursing or midwifery education [13]. Even in clinical practice, few facilities include this knowledge in their in-house education, such as clinical ladders, and personal efforts are required to acquire it. Consequently, the Japanese Nursing Association developed the Clinical Ladder of Competencies for Midwifery Practice (CLoCMiP) in 2015 [14], but training to acquire knowledge on abuse prevention was not initially incorporated.

This lack of educational opportunities [12,13] may have affected nurses’ awareness of the primary prevention of abuse. Despite the many opportunities to interact with pregnant women in obstetric care, awareness of primary abuse prevention is low, and only a few facilities in Japan are systematically working to prevent abuse in the medical field [15]. Abuse prevention in Japan is dominated by the “Hello Baby Project,” a home visit by public health nurses around the first few months after childbirth [16], which must be considered secondary or tertiary prevention.

The provision of educational opportunities for nurses on abuse prevention is expected to increase nurses’ awareness of abuse [17]. To contribute to nursing practice for abuse prevention, there is an urgent need to develop educational programs for obstetric nurses and midwives who have opportunities to care for pregnant women. The preferred learning opportunity for nurses with difficulty coordinating work hours is e-learning (a computer-based learning format) [18,19]. Some reported benefits of e-learning include flexibility, accessibility, and satisfaction [20]. As for evaluating the quality of e-learning, previous studies have demonstrated the equivalence of knowledge acquisition through in-person lectures and e-learning programs [21]. Despite the effectiveness of e-learning programs that help professionals acquire knowledge on multiple types of abuse in various countries [20,22], none have been developed in Japan. Therefore, it is necessary to create an e-learning educational program on how to deal with high-risk abuse cases and verify its effectiveness by evaluating the program’s content and participants’ degree of knowledge acquisition.

We aimed to develop an e-learning program designed to equip obstetric nurses and midwives with the essential knowledge required to effectively support and manage high-risk abuse cases. Additionally, we sought to assess the effectiveness and usefulness of this program through comprehensive evaluation. We hypothesized that obstetrics nurses and midwives would improve their post-test scores of knowledge, and the sense of efficacy and satisfaction would be improved by participating in the e-learning program we have developed.

## 2. Materials and Methods

### 2.1. Design

This study involved a single-group pre–post design with e-learning serving as the intervention. It was conducted in the obstetric wards of medical facilities near Tokyo, Japan. We conducted the study from December 2016 to July 2017. Of the 136 participants who consented to participate, 71 completed the study (pretest, e-learning educational program, and post-test; all conducted online). Data from these 71 participants were included in the analysis (52% valid responses).

### 2.2. Participants and Recruitment

The target population comprised midwives and nurses who had worked in obstetric care aged 20–65 years old. The participants were gathered through snowball sampling during the study period. Posters that invited participation in the study were displayed in the obstetric wards of medical facilities near Tokyo. The posters included a QR code to provide details of the study, which the participants accessed. Prior to their involvement in the study, the participants were provided with detailed information about the research and were given the opportunity to provide informed consent, ensuring adherence to ethical considerations. After consenting, they were asked to complete a pretest.

### 2.3. E-Learning Program

The e-learning program was developed for midwifery education by the authors (KB and YK), such as the training required to acquire the Clinical Ladder of Competencies for Midwifery Practice in Japan (CLoCMiP [14]). The e-learning program aimed to improve knowledge and awareness related to abuse prevention, such as related laws and regulations and abuse risk factors. The contents of the e-learning program, illustrated in Table 1, were based on the literature on the Japanese abuse-related social background [4,16], epidemiological data [5,6], laws and regulations [23,24], the risk factors and measurement tools for child abuse [24,25,26,27,28], and how to cooperate with organizations [29,30].

The e-learning program comprised four chapters with several sections composed of short videos (5–10 min) and a quiz with multiple-choice questions with a single correct answer. If the selected answer was correct, the user could proceed to the next section. The e-learning program took approximately 90 min to complete. The participants who completed the e-learning program were asked to complete a post-test and a satisfaction questionnaire. They were sent a reward (a Quo card worth 1000 yen) via post as an incentive.

### 2.4. Measures

Knowledge and awareness (sense of efficacy) must be increased to support and handle high-risk abuse cases [17,20,31]. As part of the e-learning educational program evaluation, we measured the following three variables: degree of knowledge, sense of efficacy, and satisfaction. These variables were measured using scales created by the authors (KB and YK). Participants’ knowledge and efficacy were measured before and after their completion of the e-learning program, while satisfaction was measured only after.

#### 2.4.1. Degree of Knowledge

To check the level of understanding of the e-learning program, we administered a 15-item knowledge test before and after the e-learning program (15 points total). The test content consisted of knowledge needed to prevent or support abuse cases, such as “definition of abuse”, “definition of specific pregnant women”, and “risk factors for abuse”.

#### 2.4.2. Sense of Efficacy

Upon the completion of the e-learning program, the participants’ sense of efficacy was evaluated through the following two questions: “If you were to encounter a case of (suspected) abuse, would you be able to respond with confidence?” and “If you were to encounter a case of (suspected) abuse, would you be able to respond positively?” All items were rated on a five-point scale (higher scores indicated higher levels of a sense of efficacy).

#### 2.4.3. Satisfaction

The quality of the e-learning program was evaluated using an anonymous questionnaire to determine the level of satisfaction in terms of “design”, “system functions”, “overall satisfaction”, “return to clinical practice”, “recommendation to colleagues”, and “content of educational materials”. All items were rated on a five-point scale (higher scores indicated higher levels of satisfaction).

### 2.5. Analysis

A *t*-test was conducted to compare the differences in the knowledge test scores (0–15 points) and the sense of efficacy (0–5 points) between the pre-test and post-test. In addition, a McNemar’s test was conducted to measure the change in the percentage of correct answers for each item between the pre-test and post-test. We calculated effect sizes [32] from the mean score differences divided by the standard deviation (SD) of the knowledge test score. Statistical analysis was conducted using SPSS version 24.0 (IBM Corp., Armonk, NY, USA), and the two-sided significance level was set at 5%. There were no missing values in this study.

### 2.6. Ethics

This research project was approved by the Ethical Committee of St. Luke’s International University (16-A075) in November 2016. The participants were provided with detailed written information about the study and offered written consent before the completion of the pretest. They were told that they could withdraw at any time without prejudice. To ensure their privacy, they were asked to complete the questionnaires online. We provided an online support system for the participants to contact the administrator (KB) to help them operate the e-learning program. To avoid potential harm [33], the researchers (KB and YK) received additional training about research ethics prior to conducting the study.

## 3. Results

### 3.1. Participant Characteristics

The participant characteristics are presented in Table 2. Regarding the participants who had previously rendered assistance in abuse cases (33.8%), almost all had responded within the organization in which they were engaged (during ward conferences, reporting to supervisors, working with medical social workers, or working with pediatrics), half of them reported the incident to the Child Guidance Center or Child and Family Support Center, and only one (1.4%) reported the incident to the police.

### 3.2. Evaluation of E-Learning

#### 3.2.1. Knowledge

Before and after the e-learning program, a pre-test and post-test were conducted to evaluate the participants’ knowledge using the total score (maximum value: 15 points) (Table 3). The scores of the pre-test and post-test ranged from 4 to 12 points (mean = 8.1 (SD 2.0)) and 7 to 15 points (mean = 11.6 (SD 1.7)), respectively). The percentages of correct answers for each of the 15 test items are listed in Table 3. A paired t-test demonstrated that the post-test scores were significantly higher (t = 14.4, *p* < 0.001). The effect size was large for the post-test (d = 1.71) [32]. McNemar’s test was conducted on the percentage of correct answers for each item in the pre-test and post-test, and statistically significant differences were found for eight items, including “What is the law that establishes the definition of abuse?” (*p* < 0.001). The item with the lowest percentage of correct answers in both the pre-test and post-test was “Matters to be considered by the Regional Council for Measures for Children in Need of Protection.”

#### 3.2.2. Sense of Efficacy

Before engaging in the e-learning program, the participants were asked the question, “If you were to encounter a case of (suspected) abuse at this time, would you be able to respond with confidence?” Out of the respondents, approximately 10% (eight individuals) answered with a rating of 4 or 5. However, following the completion of the e-learning program, the number of confident responders increased significantly to approximately 50% (35 individuals). This observed increase in confidence scores was found to be statistically significant (t = 11.0, *p* < 0.001), as illustrated in Table 4. Regarding the query, “If you were to encounter a case of (suspected) abuse at this time, would you be able to respond proactively?”, the initial responses before undertaking the e-learning program revealed that approximately 28% (20 individuals) answered with a rating of 4 or 5. However, upon completion of the e-learning program, the number of individuals who expressed proactive confidence increased significantly to around 72% (51 individuals). This observed rise in proactive response scores was found to be statistically significant (t = 8.1, *p* < 0.001), as indicated in Table 4.

#### 3.2.3. Satisfaction

Regarding the evaluation of the e-learning program, we first obtained the participants’ responses on their satisfaction with the design and system functions, overall satisfaction, whether they believed the knowledge gained could be applied to clinical practice, and whether they would recommend e-learning to colleagues (Table 5). Answering on a five-point scale, 69.0% (≥4 points) were satisfied with the design, 68.1% were satisfied with the system functions, 85.9% were satisfied overall, 76.1% believed they could apply the knowledge gained to clinical practice, and 85.9% said they would recommend e-learning to colleagues. Subsequently, the level of satisfaction with the content of the teaching materials for each section of e-learning was evaluated on a five-point scale (Table 6). For all sections, most respondents rated the content at either 4 or 5. Though approximately 90% (64/71) of the participants completed the program in less than two hours, 20% (14/71) felt the program was too long.

## 4. Discussion

### 4.1. Usefulness of the E-Learning Program

The scores on the knowledge test regarding abuse increased significantly after the implementation of e-learning (t = 14.4, *p* < 0.001) with a large effect size (d = 1.71). In previous studies [20,31], the effectiveness of e-learning for medical professionals, including midwives, social workers, and other welfare workers, has been evidenced by increased knowledge test scores, as was the case in the present study. Although it was expected that obstetric nurses would not be very familiar with the current status of abuse cases owing to their educational backgrounds [13], these results indicate, to some extent, the effectiveness of the e-learning program developed in this study.

More than 90% of the participants were positive about the operation of the e-learning program and the visibility of the screen, and there were no problems with operability. The overall satisfaction score for the program was higher than 80, and a positive evaluation was obtained for the operation method, appropriateness of the teaching materials, and satisfaction with the content. Therefore, there was a high level of satisfaction with this e-learning program, similar to previous studies [20].

According to the World Health Organization, approximately 90% of child abuse cases go unnoticed by professionals [34], which may be owing to their lack of knowledge or low self-efficacy [35]. As the participants of this e-learning program demonstrated an increase in not only knowledge but also self-efficacy, it is expected to promote awareness of and support for cases with a high risk of abuse or suspected abuse [36]. Furthermore, 86% of the respondents would recommend the developed e-learning program to their colleagues, which suggests the possibility of its further dissemination. These results indicate that an e-learning program developed to allow nurses involved in obstetrics to acquire the knowledge necessary to provide support in cases with a high risk of abuse would be useful. As previous studies [20,37,38] have reported that e-learning is useful for continuing education for nurses, there is a possibility that the e-learning program developed in this study could also become popular as a tool for continuing education in midwifery and nursing (e.g., CLoCMiP [14]).

### 4.2. Improvements in E-Learning Programs for Dissemination

The observed significant increase in the percentage of accurate responses to all three items, namely the most common types of abuse, the percentage of female victims of domestic violence, and the scope of the Regional Council for Measures for Children in Need of Protection, following the completion of the course, suggests that the e-learning program effectively facilitated the acquisition of specific content knowledge among the participants. This e-learning program may be one way to overcome one of the causes of abuse deaths in Japan: the lack of knowledge about the Regional Councils for Measures for Children in Need of Protection and the lack of cooperation caused by the lack of knowledge [39]. Meanwhile, the items for which the percentage of correct answers did not increase significantly after the program may have been difficult for the participants to understand and thus must be corrected before dissemination.

While approximately 90% of the participants completed the program in less than two hours, 20% felt it was too long. One factor may be that this e-learning program was somewhat longer than another e-learning program, which took 45 min to complete [40]. Therefore, the content to be included in the program needs to be more concise. Most of the seven items for which the percentage of correct answers did not increase significantly after the program were answered correctly in the pretest, which suggests that students may have acquired the knowledge before the program. Considering the test results obtained in this study, there is a possibility of condensing the content related to the aforementioned items. It is crucial to provide explanations, supported by statistical data, that depict the actual situation of abuse cases (primarily derived from Sections 1-1, 2-2, and 2-3 of the e-learning program, Table 3). On the other hand, the content pertaining to abuse risk factors (largely covered in Sections 1-2 and 2-1 of the e-learning program), which the participants were already acquainted with prior to the program, can be reduced. In addition, in a previous study [41] that conducted an e-learning program with a similar duration to that of the present study, the percentage of respondents who answered “too long” was almost the same; therefore, we believe that the percentage of respondents in our study who answered “too long” was appropriate. The results suggest that the e-learning system developed in this study could be widely used in Japan by implementing the suggested improvements.

### 4.3. Limitations

This study had several limitations. First, it did not have a control group that did not receive e-learning education. There is no way to fully attribute whether the pretesting process influenced the results; thus, external validity could be affected. Given that the authors utilized their own measures of effectiveness, it is important to acknowledge the possibility that the pretest experience may have influenced the learning effect. To definitively establish the study’s contribution to enhancing knowledge, it would be necessary to revalidate the findings through a comparative study design incorporating a control group for comparison. As this study was limited to a short-term evaluation of knowledge acquisition after participation in an e-learning program, a long-term evaluation of knowledge maintenance is needed.

Second, as this study targeted midwives and nurses working at obstetric institutions in Tokyo, the results cannot be generalized to all nurses in Japan. Furthermore, because more than 90% of the participants were midwives, a larger sample, including a significant number of nursing professionals, is required.

Third, of the 108 participants who consented to engage in the study, 51 did not complete the educational program (47% dropout rate). Of these, approximately half were unable to access the educational program after completing the pretest. The researchers sent reminder e-mails to the registered e-mail addresses of each participant two weeks before the end of the study period, informing them about where to access the e-learning program; they also responded to participants’ system-related problems individually by e-mail. However, despite these measures, there were still dropouts. As most of the participants who did not complete the educational program did not use e-learning, it is possible that there was a need for a face-to-face version of the program. Furthermore, as a previous study [42] demonstrated that face-to-face and e-learning are equally effective, it may be useful to create both e-learning and face-to-face educational programs in the future so that participants can choose based on their preferences.

Fourth, this study was conducted before the COVID-19 pandemic and did not consider the long-term impacts on society and health caused by COVID-19 [43]. There is a need to revise the e-learning program to incorporate content that focuses on identifying and enhancing support for mothers who are at high risk of abuse (e.g., vulnerable to pandemic-related stressors) [44].

Despite these limitations, this is the first study to develop an e-learning program that efficiently provides the necessary multifaceted knowledge regarding the primary prevention of abuse cases and to confirm the effectiveness of the “knowledge,” “efficacy,” and “satisfaction” improvements obtained in this study. Future research should modify the content of the e-learning program and test the effectiveness of the program with professionals not limited to the nursing profession or the medical frame of reference. To truly prevent abuse, the dissemination of abuse e-learning programs that have been validated for effectiveness is necessary.

## 5. Conclusions

We demonstrated the usefulness of the developed e-learning program for acquiring knowledge and improving the sense of efficacy necessary to support high-risk cases of abuse. To our knowledge, this is the first study in Japan to develop and demonstrate the effectiveness of the e-learning program for obstetric nurses and midwives to acquire the knowledge to prevent abuse. In subsequent studies, the contents need to be revised based on issues identified in this study, and their effectiveness should be tested using a comparative study design. However, this e-learning tool could be used from the perspective that abuse prevention is an urgent need in the perinatal nursing field. The practical use of one of the training programs in CLoCMiP [14], the national midwifery clinical ladder in Japan, may contribute to the primary prevention of abuse among midwives who are most often involved in the support of pregnant women during prenatal checkups.

## Figures and Tables

**Table 1 ijerph-20-06317-t001:** Overview of e-learning.

Chapter	Section	Time	Objectives
1. Basic knowledge of abuse response	1. Increase in abuse and background	9 min	Explain the definition of abuse. Explain the background of and increase in abuse.
	2. Response to abuse and related laws	13 min	Explain how to respond to abuse based on the law. Visualize the response to abuse.
2. Basic knowledge of abuse prevention	1. Risk factors for child abuse	11 min	Discuss risk factors for abuse. Describe specific pregnant women.
	2. Perinatal mental health	13 min	Describe bonding disorders. Describe maternity blues and postpartum depression. Describe the prevention and response to bonding disorders. Describe how to deal with maternity blues and postpartum depression.
	3. Domestic violence	20 min	Explain the concept of DV, its risk factors, and health implications. Explain the DV screening scale and methods. Explain how to respond to a positive screening.
3. Basic knowledge necessary for cooperation with related organizations	1. Cooperation with related organizations	7 min	Discuss the necessity of cooperation with related organizations for the prevention of child abuse. Introduce the Regional Council for Measures for Children in Need of Protection.
4. Roleplay	1. Communication in the medical field 1	5 min	Imagine and practice cooperation among staff members when encountering a pregnant woman suspected of abuse in an obstetric outpatient clinic.
	2. Communication in the medical field 2	12 min	Visualize support for pregnant women who have been subjected to DV and apply it in practice.

Note. DV = domestic violence.

**Table 2 ijerph-20-06317-t002:** Participants’ characteristics (N = 71).

Variable	Response (n)	Mean (SD) or n (%)
Age, years, range 23–61	69	36.2 (9.6)
Experience in OB, years, range 0–34	69	9.6 (7.3)
Nursing education	71	
4-year university		30 (42.2)
Vocational school		18 (25.4)
Graduate school		14 (19.7)
Junior college		9 (12.7)
Occupation	71	
Midwife		70 (98.6)
Nurse		1 (1.4)
Midwife education	70	
4-year university		19 (26.8)
Vocational school		22 (31.0)
Graduate school		15 (21.1)
Junior college		14 (19.7)
Experience of receiving education related to support for abuse cases	71	
Yes		56 (78.9)
No		15 (21.1)
Clinical experience	71	
Perinatal care with tertiary emergency		32 (45.1)
Perinatal care with secondary emergency		13 (18.3)
Other perinatal care		26 (36.6)
Case handling experience	71	
Never		35 (49.3)
Encountered suspected abusive cases but could not provide support		12 (16.9)
Provided support in such cases		24 (33.8)

Note. OB = obstetric care; SD = standard deviation. Education related to support for abuse cases = participants’ experience of voluntary participation in workshops, enrolling in programs within the curriculum of midwifery or nursing education, and receiving education within their institutions.

**Table 3 ijerph-20-06317-t003:** Percentage of correct answers for each knowledge test item.

Knowledge Test Item Contents (Related Chapter Sections)	Percentage of Correct Answers n (%)		Difference in Percentages of Correct Answers	*p*-Value
	Pretest	Posttest		
1. What is the law that establishes the definition of abuse? (1-1)	40 (56.3)	63 (88.7)	32.4	<0.001
2. How old is a child in the definition of child abuse? (1-1)	50 (70.4)	67 (94.4)	24.0	<0.001
3. What are the nurturing attitudes that fall under the definition of child abuse? (1-1)	38 (53.5)	59 (83.1)	29.6	<0.001
4. What is the most commonly reported abuse in recent years? (1-1)	9 (12.7)	47 (66.2)	53.5	<0.001
5. Breakdown of abuse deaths. (1-1)	33 (46.5)	59 (83.1)	36.6	<0.001
6. Obligation of the person who discovers abuse to notify. (1-2)	69 (97.2)	71 (100)	2.8	-
7. Responding to and supporting abuse cases. (1-2)	55 (77.5)	59 (83.1)	5.6	0.481
8. Definition of specific pregnant women. (2-1)	60 (84.5)	71 (100)	15.5	-
9. Risk factors for child abuse. (2-1)	65 (91.5)	68 (95.8)	4.3	0.508
10. Bonding failure. (2-2)	67 (94.4)	70 (98.6)	4.2	0.375
11. Definition of postpartum depression. (2-2)	24 (33.8)	47 (66.2)	32.4	<0.001
12. Percentage of female victims of DV (2-3)	14 (19.7)	67 (94.4)	74.7	<0.001
13. Provisions set forth in the DV Prevention Law (2-3)	49 (69.0)	57 (80.3)	11.3	0.096
14. Scope of the Regional Council for Measures for Children in Need of Protection. (3-1)	7 (9.9)	35 (49.3)	39.4	<0.001
15. Matters to be considered by the Regional Council of Countermeasures for Children Requiring Aid. (3-1)	35 (49.3)	43 (60.6)	11.3	0.115

Note. N = 71, McNemar’s test. DV = domestic violence.

**Table 4 ijerph-20-06317-t004:** Changes in knowledge test scores and participants’ sense of efficacy.

	Pretest		Posttest		Mean Difference	SD of Mean Difference	
	Average	SD	Average	SD			t Value
Knowledge score	8.1	2.0	11.6	1.7	3.5	2.1	14.4 ***
Participants’ sense of efficacy (confidence)	2.1	1.0	3.5	0.8	1.4	1.1	11.0 ***
Participants’ sense of efficacy (proactiveness)	2.9	1.2	3.8	0.8	1.0	1.0	8.1 ***

Note. N = 71 *** *p* < 0.001. SD = standard deviation.

**Table 5 ijerph-20-06317-t005:** Evaluation of the e-learning program (participant satisfaction).

Evaluation Items	Degree of Satisfaction n (%)				
	1 (Low)	2	3	4	5 (High)
Design of e-learning	0	2 (2.8)	20 (28.2)	22 (31.0)	27 (38.0)
System functions of e-learning	0	4 (5.6)	19 (26.8)	26 (37.1)	22 (31.0)
Overall satisfaction with e-learning	0	1 (1.4)	9 (12.7)	39 (54.9)	22 (31.0)
Whether the knowledge gained from e-learning can be applied to future clinical practice	0	2 (2.8)	15 (21.1)	36 (50.7)	18 (25.4)
Whether they would recommend e-learning to their colleagues	0	1 (1.4)	9 (12.7)	26 (36.6)	35 (49.3)

Note. N = 71.

**Table 6 ijerph-20-06317-t006:** Evaluation of the e-learning program (satisfaction with the content of the educational materials).

Chapter	Section	Degree of Satisfaction n (%)				
		1 (Low)	2	3	4	5 (High)
1. Basic knowledge of abuse response	1	0	2 (2.8)	11 (15.5)	34 (47.9)	24 (33.8)
	2	0	2 (2.8)	12 (16.9)	35 (49.3)	22 (31.0)
2. Basic knowledge of abuse prevention	1	0	1 (1.4)	10 (14.1)	31 (43.7)	29 (40.8)
	2	0	1 (1.4)	12 (16.9)	34 (47.9)	24 (33.8)
	3	0	2 (2.8)	12 (16.9)	32 (45.1)	25 (35.2)
3. Basic knowledge necessary for cooperation with related organizations	1	0	3 (4.2)	14 (19.7)	34 (47.9)	20 (28.2)
4. Roleplay		0	6 (8.5)	14 (19.7)	23 (32.4)	28 (39.4)

Note. N = 71.

## Data Availability

The data presented in this study are available from the corresponding author upon request.

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
