# Peer review of "An E-Learning Program for Continuing Midwifery Education on Handling High-Risk Abuse Cases: A Pretest–Posttest Design"

_ijerph, 2023, doi:10.3390/ijerph20136317_

Round 1

Reviewer 1 Report

1. Abstract should be stated clearer for highlighting the contributions.

2. Conclusions should be stated clearer for highlighting the contributions. In addition, please mention the implications or suggestions.

3. Data processing procedure needs to be stated clearer.

4. The manuscript should be proof read, otherwise it is difficult to understand and read. Please re-check the grammar and spelling.

Reviewer 2 Report

DESIGN: the description must be implemented

PROCEDURE: When referring to studies, the bibliographical references must be inserted. The procedure is not described in sufficient detail and does not respect scientific rigour

CONCLUSIONS: a broader development of the conclusions is missing. The limitations of the study and more structured research implications must be included

Reviewer 3 Report

data source is old data

Reviewer 4 Report

First of all, I would like to thank the authors for their contribution "An E-Learning Program for Continuing Midwifery Education 2 on Handling High-Risk Abuse Cases: A Pretest–Posttest Design".

Introduce the paper describing what the paper is about. Expand to emphasize the problem leading to a clear set of research questions and objectives the research addresses.

The introduction can be strengthened further by elaborating on the research gap that this paper is attempting to close, i.e., why there have been limited studies to-date by citing prior literature, and why is this study important. 

I would like to know more about the Japanese educational system.

I would like to know more about the ethical issues of the study, Author(s) need to mention ethical issues for their study and the relations between science and society. I propose to add the following reference in the sub-section 2.4 Ethics:

Petousi, V., & Sifaki, E. (2020). Contextualizing harm in the framework of research misconduct. Findings from discourse analysis of scientific publications, International Journal of Sustainable Development, 23(3/4), 149-174, DOI: 10.1504/IJSD.2020.10037655

Conclusions are rather short.

I wish you the best of luck with the revisions of your manuscript.

Author Response

Please see the attacment.

Reviewer 5 Report

Thank you for the opportunity to review your paper. I found that it was well written. As an instructor of child language and developmental psychology, I find the content of this paper to be of high importance, especially when considering pre- and post-natal trauma and the effects on cognitive development. 

You've chosen to do the correct statistical analysis, but it could be strengthened if followed up with a Cohen´s d for the effect size. You also fully addressed the limitations of the study. 

Author Response

Please see tha attachment.

Round 2

Reviewer 4 Report

First of all I would like to thank the authors for the revised version of their work. It's still missing the following reference about the ethical issues.I propose to add the following reference in the sub-section 2.4 Ethics:

Petousi, V., & Sifaki, E. (2020). Contextualizing harm in the framework of research misconduct. Findings from discourse analysis of scientific publications, International Journal of Sustainable Development, 23(3/4), 149-174, DOI: 10.1504/IJSD.2020.10037655

Author Response

Thank you very much for your kind attention again. We have cited the literature as you suggested. We hope this adequately addresses your concern.

P4, lines144-152

2.6. Ethics

This research project was approved by the Ethical Committee of St. Luke’s International University (16-A075) in November 2016. Participants were provided with detailed written information about the study and offered written consent before the completion of the pretest. They were told that they could withdraw at any time without prejudice. To ensure their privacy, they were asked to complete the questionnaires online. We provided an online support system for participants to contact the administrator (KB) to help them operate the e-learning program. To avoid potential harm [33], researchers (KB and YK) received additional training about research ethics prior to conducting the study.

The following paper has been added.

  1. Petousi, V; Sifaki, E. Contextualizing harm in the framework of research misconduct. Findings from discourse analysis of scientific publications, Int J Sustainable Development 2020, 23, 149-74. DOI: 10.1504/IJSD.2020.10037655